# Comparison of Enhanced Photocatalytic Degradation Efficiency and Toxicity Evaluations of CeO_2_ Nanoparticles Synthesized Through Double-Modulation

**DOI:** 10.3390/nano10081543

**Published:** 2020-08-06

**Authors:** Jang Hyun Choi, Jung-A Hong, Ye Rim Son, Jian Wang, Hyun Sung Kim, Hansol Lee, Hangil Lee

**Affiliations:** 1Department of Biological Sciences, College of Natural Science, Inha University, 100 Inha-ro, Michuhol-gu, Incheon 22212, Korea; jhchoi3@inha.edu; 2Department of Chemistry, Sookmyung Women’s University, Seoul 04310, Korea; junga4540@naver.com; 3Department of Chemistry, Pukyong National University, Busan 48513, Korea; syl75218@daum.net; 4Canadian Light Source and University of Saskatchewan, 44 Innovation Boulevard, Saskatoon, SK S7N 2 V3, Canada; Jian.Wang@lightsource.ca

**Keywords:** modified CeO_2_ NPs, oxygen vacancy, photocatalytic degradation, scanning transmission X-ray microscopy, toxicity, cell penetration

## Abstract

We demonstrated that Fe/Cr doped and pH-modified CeO_2_ nanoparticles (NPs) exhibit enhanced photocatalytic performance as compared to bare CeO_2_ NPs, using photocatalytic degradation. To assess the toxicity level of these double-modified CeO_2_ NPs on the human skin, they were introduced into HaCaT cells. The results of our conventional cellular toxicity assays (neutral red uptake and 3-(4,5-dimethylthiazol-2-yl)-2,5-diphenyltetrazolium bromide for assays) indicated that Cr@CeO_x_ NPs prompt severe negative effects on the viability of human cells. Moreover, the results obtained by scanning transmission X-ray microscopy and bio-transmission electron microscope analysis showed that most of the NPs were localized outside the nucleus of the cells. Thus, serious genetic toxicity was unlikely. Overall, this study highlights the need to prevent the development of Cr@CeO_x_ NP toxicity. Moreover, further research should aim to improve the photocatalytic properties and activity of these NPs while accounting for their stability issues.

## 1. Introduction

Versatile metal oxide nanoparticles (MO NPs) have been extensively applied in various scientific and engineering fields such as electrochemical reactions, photocatalytic degradation, fuel cells, and photoreactions [1,2,3,4,5,6,7,8,9]. However, nanometer-sized MO NPs can easily penetrate the human body through the skin, and, thus, the biocompatibility of these nanoparticles (NPs) should be seriously considered as a safety measure [10,11,12,13]. Moreover, they may be detrimental to human health, which is potentially toxic to cells. Therefore, their toxicity levels must be clearly monitored before their employment in various applications, especially those involving contact with human bodies [14,15,16]. Hence, it is important to not only enhance the photocatalytic activities of the MO NPs through various modifications but also ensure their safety for humans (e.g., through toxicity evaluations) [17,18,19].

CeO_2_ NPs are among the well-known, key components of photocatalysts (such as TiO_2_, ZnO, WO_3_, etc.) and have also garnered interest in the field of biomedicine [20,21]. Double modulating strategies involving the doping of CeO_2_ NPs with a transition metal and their pH modification under basic conditions (TM@CeO_x_ NPs) have been reported to be particularly effective in enhancing the photocatalytic activity of CeO_2_ NPs [22,23,24]. These modifications create oxygen vacancies (O_V_) on the surface of CeO_2_ NPs, which can act as adsorption and active sites [25]. Therefore, through the double modification of CeO_2_ NPs, we can enhance the photocatalytic activity of CeO_2_ NPs. The promising characteristics of CeO_2_ NPs have a direct impact on human health and the environment. They are very small (<100 nm) and can easily enter human tissues through the skin or by following prolonged exposure, which possibly causes severe damage [26]. Several studies have reported that CeO_2_ NPs can have significant cytotoxic effects in a variety of living organisms. Such effects are influenced by NP concentration, exposure time, and pH conditions [27,28,29]. In addition, CeO_2_ NPs have been shown to display toxic effects associated with the level of reactive oxygen species (ROS) both in vivo and in vitro [17,30]. More specifically, if CeO_2_ NPs of this type penetrate the nucleus containing DNA information, they can cause severe problems, including chromosomal abnormalities [31,32].

Herein, we aimed to optimize the photocatalytic properties of CeO_2_ NPs by doping them with Cr or Fe ions and treating the samples under basic conditions (pH = 13.5), which are known to enhance the photocatalytic properties. The photocatalytic activities of the modified CeO_2_ NPs were assessed by testing the degradation of 4-chlorophenol (4-CP), 2,4-dichlorophenol (DCP), and HCOOH in aqueous solutions. Then, we quantified the change in HaCaT cell viability in the presence of Cr(or Fe)@CeO_x_ NPs and monitored the toxic effects of the NPs on human skin. The toxicity results obtained in this study can be used to evaluate the potential of Cr(or Fe)@CeO_x_ NPs for applications in conventional chemical reactions and emerging biomedical research.

## 2. Materials and Methods

### 2.1. Sample Preparation of TM@CeO_x_ NPs

We first synthesized CeO_2_ NPs using a modified thermal method [33]. Afterward, we prepared 1 mol% TM@CeO_x_ NPs using TM(NO_3_)_3_∙9H_2_O (99%) (TM = Cr or Fe) and pH treatment under basic conditions (pH = 13.5) using KOH and then maintained for 30 min. After confirming sol-gel solutions, we transferred these solutions to Teflon-lined autoclave reactors and then sealed and heated at 220 °C for 7 h in a convection oven. The final products were named Cr@CeO_x_ and Fe@CeO_x_ NPs. Lastly, the samples were filtered and washed with double-distilled water (DDW) to remove any residues. All substances were purchased from Sigma-Aldrich (St. Louis, MO, USA).

### 2.2. Introduction of TM@CeO_x_ NPs into Cells and Sample Preparation

The HaCaT cells were kindly provided by Dr. Kwon (Inha University, Korea). HaCaT cells were grown in 100-mm culture dishes (Corning, Corning, NY, USA) until 20% confluence and exposed to 100 μg/mL of three samples (i.e., CeO_2_, Cr@CeO_x_, or Fe@CeO_x_) for 24 h. After incubation, the cells were harvested with 0.25% trypsin-EDTA (GIBCO, Grand Island, NY, USA) and then fixed in freshly prepared 4% paraformaldehyde (EMS, Hatfield, PA, USA), 1% glutaraldehyde (Merck, Darmstadt, Germany), and 0.1 M Sorensen’s phosphate buffer for 2 h. Next, the cells were post-fixed with 1% osmium tetroxide (Sigma-Aldrich, St. Louis, MO, USA) and 0.1 M Sorensen’s phosphate buffer for 30 min and then washed three times with 0.1 M Sorensen’s phosphate buffer. After infiltration and embedment, the samples were cut with a diamond knife (Diatome, Biel, Switzerland) into sections of ~60 nm in thickness on an Ultracut E ultramicrotome (Reichert-Jung, Vienna, Austria) and collected on a transmission electron microscope (TEM) grid (EMS, Hatfield, PA, USA). Lastly, the sections were stained with uranyl acetate (Sigma-Aldrich, St. Louis, MO, USA).

### 2.3. Cell Viability Assays

The cytotoxic effects of CeO_2_ and TM@CeO_x_ NPs were assessed by 3-(4,5-dimethylthiazol-2-yl)-2,5-diphenyltetrazolium bromide (MTT) and neutral red uptake (NRU) assays, as described in previous works [34]. In brief, 2 × 10^4^ HaCaT cells per well were seeded in 96-well cell culture plates (Corning, Corning, NY, USA) and exposed to TM@CeO_x_ NPs or CeO_2_ NPs for 72 h. For the MTT assays, the cells were washed twice with phosphate-buffered saline (PBS, GIBCO, Grand Island, NY, USA) and exposed to NPs. Lastly, an MTT solution (Sigma-Aldrich, St. Louis, MO, USA) was added to each well at a final concentration of 0.5 mg/mL and the cells were incubated for 1 h. The formazan crystals that formed during the process were dissolved in 50% dimethyl sulfoxide (DMSO) (Sigma-Aldrich, St. Louis, MO, USA) and 50% methanol (Merck, Darmstadt, Germany). For the NRU assays, the NP-exposed cells were washed twice with PBS and incubated for 4 h in OPTI-MEMI (GIBCO, Grand Island, NY, USA), which contained 40 ng/mL of the neutral red reagent (Sigma-Aldrich, St. Louis, MO, USA). After incubation, the wells were eluted with 50% ethanol (Merck, Darmstadt, Germany) and 1% glacial acetic acid (Merck, Darmstadt, Germany). The absorbance of the plates was measured using an XFluor4 microplate reader (TECAN Japan, Kawasaki, Japan) at 595 nm and 540 nm in the case of the MTT and NRU assays, respectively.

### 2.4. Photocatalytic Measurements

We used 4-chlorophenol (4-CP, Sigma-Aldrich, ≥99%), 2,4-dichlorophenol (2,4-DCP, Sigma-Aldrich, ≥99%), and formic acid (Sigma-Aldrich, ≥96%) as target contaminants. The suspensions were magnetically stirred in the dark for 10 min to establish the absorption-desorption equilibrium. The three different CeO_2_ NPs were dispersed in distilled water (0.5 g L^−1^) under sonication. The aqueous suspension was stirred for 15 min to allow the equilibrium adsorption of substrates on CeO_2_. A 300-W Xe arc lamp (Oriel) combined with a 10-cm IR water filter and a cut-off filter (λ > 320 nm for ultraviolet (UV) light) was used as a light source. A typical incident light intensity was determined to be about 100 mW cm^−2^ in the wavelength range 320 nm and stirred magnetically during irradiation (rotation per min (rpm) is maintained at 80). Sample aliquots were withdrawn by a 1-mL syringe intermittently during the photoreaction and filtered through a 0.45-μm PTFE filter (Millipore) to remove suspended CeO_2_ NPs. The changes in the concentrations of 4-chlorophenol and 2,4-dichlorophenol were measured by high-performance liquid chromatography (HPLC, Shimadzu UFLC LC-20AD pump). To examine the variations in formic acid concentration, we employed an ion chromatography (IC, Thermo) conductivity detector combined with a Dionex IonPac AS 22 (4 mm × 150 mm) column.

### 2.5. Characterizations

The morphologies of the samples were characterized by bio-TEM (FEI Talos L120C) at an acceleration voltage of 120 kV. The X-ray diffraction (XRD) patterns of the bare CeO_2_ and the two modified CeO_2_ NPs were obtained by applying Ni-filtered Cu-K*α* radiation using a Rigaku D/Max-A diffractometer. Scanning transmission X-ray microscopy (STXM) analysis was performed at the 10ID-1 Soft X-ray spectro-microscopy (SM) beamline of the Canadian Light Source (CLS). Image stacks were acquired by X-ray absorption spectroscopy (XAS) to extract the Ce *M*-edge and O *K*-edge spectra. Raman spectra data were obtained with an Ar^+^ ion laser (Spectra-Physics Stabilite 2017; λ_ex_ = 514.5 nm) excitation source connected to a Horiba Jobin Yvon TRIAX 550 spectrometer.

## 3. Results

### 3.1. Characterization of TM@CeO_2_ NPs

First, the XRD patterns of CeO_2_ and the Cr(or Fe)@CeO_x_ NPs were used to identify changes in the crystal structure (Figure 1a). As expected, all the XRD patterns were typical of the fluorite-structured CeO_2_ without any clear structural changes. We observed peaks at 2*θ* = 28.7°, 33.2°, 47.7°, 56.5°, and 59.2°, which correspond to the (111), (200), (220), (311), and (222) reflections, respectively (JCPDS card No. 41-1455). The lattice constant calculated for CeO_2_ was 5.41 ± 0.03 Å, which is the same as those calculated for the two double-modified CeO_2_ NPs. Thus, all three samples exhibit the same structure. However, to evaluate a more accurate structural change, it is necessary to check the intensity and broadening changes of the XRD peaks because these changes can cause slight changes in sample size and structure. For this purpose, we closely observed the changes in the peaks of the (111) and (200) reflections of the sample displayed in Figure 1a. As a result, no peak intensity change or peak broadening change was observed between the three samples. Hence, this result confirms that there were no major structural changes among the three tested samples.

Figure 1b shows the Raman spectra of CeO_2_, Cr@CeO_x_, and Fe@CeO_x_ NPs. All samples were characterized by a prominent band at ∼464 cm^−1^, which corresponds to the F_2g_ Raman active mode in the cubic fluorite structure of CeO_2_ [35,36]. The main F_2g_ band shifted to a slightly lower wavenumber (~5 cm^–1^) in the Cr@CeO_x_ and Fe@CeO_x_ NPs when compared to that of the CeO_2_ NPs. This shift was caused by the incorporation of Cr and Fe ions in the fluorite lattice (during the doping process). Additionally, the Raman spectrum of Fe@CeO_x_ was characterized by a weak band at 412 cm^−1^ (blue mark), which can be assigned to hematite (α-Fe_2_O_3_) [35]. This result suggests that α-Fe_2_O_3_ was formed on the surface as a secondary phase. In the case of Cr@CeO_x_, a weak band corresponding to Cr_2_O_3_ was observed at ∼650 cm^–1^ (red mark) [36]. Notably, there were no significant differences between the three kinds of CeO_2_ NPs. The synthesized NPs did not present large structural differences. Therefore, they were expected to have similar photocatalytic and cytotoxicity properties under the same conditions.

The local electronic structures in the unoccupied state regions of the metal-doped and pH-modified (basic conditions, pH = 13.5) CeO_2_ NPs were also compared based on the results of XAS analyses shown in Figure 1c [37,38]. It is also clear that the spectral features of bare CeO_2_ NPs are very different from those of the modified CeO_2_ NPs. In detail, this strengthened our view of the different electronic structures of the critical phases of Cr@CeO_x_ and Fe@CeO_x_ NPs, which is modified from the bare CeO_2_ NPs.

To confirm this, the O *K*-edge spectra of the CeO_2_ NPs were compared and a large change was observed among the three different CeO_2_ NPs unlike the XRD and Raman measurements, which had no structural change. Due to the large overlap of the O 2*p* and Ce 4*f* wave functions, the pre-edge peak (peak A) at the O *K*-edge yields the density of Ce 4*f* states in CeO_2_ NPs [39,40,41]. In other words, the O *K*-edge spectra (Figure 1c) confirmed that the intensity of peak A for the Cr@CeO_x_ and Fe@CeO_x_ NPs was smaller than that for bare CeO_2_ NPs. To evaluate a more accurate change, the intensity ratio between peak A and the eg orbital was calculated for the three samples. The calculated values were 0.881 ± 0.04, 0.394 ± 0.02, and 0.524 ± 0.03 for CeO_2_, Cr@CeO_x_, and Fe@CeO_x_, respectively. As mentioned, it can be seen that the samples obtained by the double modulation (Cr@CeO_x_ and Fe@CeO_x_ NPs) show a larger change than the bare CeO_2_ NPs. From this calculated result, it is expected that the number of oxygen vacancies can be directly correlated to the photocatalytic degradation efficiency (see Figure 2).

Lastly, we obtained TEM images of the two modified CeO_2_ NPs to determine the particle size compared with that of the bare CeO_2_ NPs, which are known to affect cellular uptake and cytotoxicity (see Figure 1d). These TEM images revealed that all the NPs possessed diameters of 10–60 nm (average size is ~20 ± 3 nm), which are small enough to enter cells via micropinocytosis. The histogram in Figure 1e represents the statistical size distribution including the average size and their deviation. The average particle size of the three NPs were estimated 20.36 ± 3.16 nm, 20.78 ± 4.47 nm, and 22.90 ± 3.74 nm for CeO_2_, Cr@CeO_x_, and Fe@CeO_x_ NPs. Furthermore, using the Scherrer equation, the average particle size calculated was 22.53 nm, 21.40 nm, and 21.40 nm. These calculated values were quietly well matched with those obtained by TEM analysis [42].

Photocatalytic degradation and cell stability experiments were performed based on the structure and electronic structure change information shown in Figure 1.

### 3.2. Photocatalytic Degradation Activity Measurements

To reveal the role of the O_V_ (through Cr or Fe doping and the basic treatment at pH = 13.5) on CeO_2_ NPs, we compared the photocatalytic degradation (PCD) rates of three different organic pollutants (i.e., (a) 4-CP, (b) 2,4-DCP, and (c) HCOOH) as representative organic compounds of aromatic (i.e., 4-CP and 2,4-DCP) and aliphatic (HCOOH) compounds in the presence of CeO_2_ [43,44,45]. As shown in Figure 2, only the phenolic compounds (i.e., (a) 4-CP and (b) 2,4-DCP) were clearly degraded in all the NPs (CeO_2_, Cr@CeO_x_, and Fe@CeO_x_), while formic acid was not degraded in all. In detail, the photocatalytic activity of the Cr@CeO_x_ NPs was superior (i.e., high PCD rates) than that of the bare CeO_2_ or Fe@CeO_x_ NPs due to the enhanced number of O_V_ resulting from the transition metal doping and pH treatment, which we confirmed in Figure 1c.

The PCD mechanisms of the phenolic compounds were investigated by performing the reaction in the presence of various probing reagents. In other words, this phenomenon of oxygen vacancy induced complexation enables additional UV light absorption through ligand-to-metal charge transfer (LMCT) reactions between the absorbed molecules (ligand including phenol compounds) and the Ce defect site on the surface. Our results are very similar to those of the LMCT between the phenolic compounds and Ti(IV) on TiO_2_ [46,47]. The tested pollutant molecules are likely adsorbed onto the surface of the NPs through a phenolate linkage (Equation (1)).
(1)$$\mathrm{TM}@\mathrm{CeOx}-\mathrm{OH}+\mathrm{HO}-\mathrm{Ph}\;\to \mathrm{TM}@\mathrm{CeOx}-O-\mathrm{Ph}+H_{2}O$$

The surface complexation of Equation (1) is essentially a condensation reaction between a surface hydroxyl group on the TM@CeO_x_ NPs and an adsorbed hydroxyl group. On the other hand, aliphatic compounds do not seem to form such complexes in aqueous TM@CeO_x_ suspensions, judging from the fact that the addition of excess methanol did not inhibit the degradation of 4-CP or 2,4-DCP. As a result, only the phenol molecules among the three tested molecules can be explained by the photolysis reactions in our experiments.

The intensity changes between the two peaks (peaks A and e_g_) shown in the XAS data (see Figure 1c) were closely related to the number of O_V_. We also know that PCD efficiency is related to O_V_. Therefore, the relationship between the intensity change shown in the XAS data and the PCD efficiency should also be evaluated. To confirm this relationship, the decomposition rates of 4-CP and 2,4-DCP molecules obtained after 1 h were compared. The PCD efficiency of the two organic molecules for the three tested samples (CeO_2_, Cr@CeO_x_, and Fe@CeO_x_) was measured to be 0.774:0.435:0.523 (4-CP) and 0.673:0.378:0.460 (2,4-DCP), respectively. Furthermore, it was confirmed that these PCD efficiencies were somewhat similar to the intensity ratio obtained from XAS. From these results, we can see that the PCD efficiency can be predicted through the O_V_ obtained by using XAS.

### 3.3. STXM Measurements

Thus, we analyzed the structure, the changes in the electronic structure, and the photocatalytic properties of the two doped samples. Subsequently, we evaluate the toxicity and cell permeation of the three samples, as described above.

The modified CeO_2_ NPs that penetrated the HaCaT cells after their incubation were first analyzed by STXM to determine their intracellular loci (as either peripheral or central, including the nucleus regions inside the cells). To study the permeation of the CeO_2_ NPs into the cell, stack images were obtained at a photon energy of 884.1 eV corresponding to the Ce M_5_-edge. To clarify the presence of the NPs, we evaluated the change in intensity of the NPs depending on the location of the Ce M-edge. Appendix A shows a change in the intensity of Cr@CeO_x_ NPs according to the fixed photon energy (see Supporting Information). The modified CeO_2_ NPs are shown in black in Figure 3a,b. As can be seen in the two STXM images obtained at a photon energy of 884.1 eV, most of the permeated NPs were distributed on both sides of the cell membrane.

Another concern is the change in the electronic structure of the NPs that penetrate the cells. Cell viability experiments have shown that Cr@CeO_x_ NPs are severely toxic in cells, possibly due to changes in their electronic structure. Hence, the same sample areas of the STXM XAS spectra (see the Ce M-edge spectra in Figure 3c,d were also analyzed. The XAS spectra of the modified CeO_2_ NPs (Cr@CeO_x_ and Fe@CeO_x_ NPs) were measured, and the peaks corresponding to M_5_ and M_4_ are clearly seen. In addition, additional satellite peaks such as Y’ and Y, which originate from the transitions to the 4*f* states in the conduction band, are also observed for CeO_2_. It is known that the bare CeO_2_ NPs show two major peaks at ~883.7 eV and ~901.3 eV for the Ce M_5_-edge and M_4_-edge, respectively. In addition, there are two post edge peaks at ~889.1 eV (Y’) and 906.7 eV (Y), respectively [43,44].

To understand the impact of the changes in the electronic structure of the NPs, Ce M_5,4_ -edge spectra were collected from the Cr/Fe doped CeO_2_ NPs. As shown in Figure 3d, it can be seen that Fe@CeO_x_ NPs exhibited almost the same photon energy (M_5_-edge: 884.1 eV and M_4_-edge: 901.4 eV) and shape as that of the bare CeO_2_ NPs. Thus, these NPs penetrate the cell without interacting with various components in the cell. On the other hand, large changes in the electronic structure of the Cr@CeO_x_ NPs (M_5_-edge, 885.9 eV and M_4_-edge: 902.3 eV) were observed when compared with those of Fe@CeO_x_ NPs. As can be seen in Figure 3c, in particular, the *M*_4_ and Y peaks show significant differences from the electronic structure of the conventional CeO_2_ NPs. In other words, when these NPs penetrate the cells, they react with substances in the cells, which causes changes in the electronic structure. As a result, this change in electronic structure could be used as a measure of toxicity.

In summary, these data demonstrate the existence of different electronic structures for the Fe@CeO_x_ and Cr@CeO_x_ NPs. The former presented the typical electronic structure of CeO_2_ NPs, whereas the latter reacted with the materials present in the cell, which is indicated by a change in the electronic structure from that of CeO_2_. Notably, none of the NPs penetrated the nucleus (center of the cell). We interpret the electronic structure change seen for the Cr@CeO_x_ NPs as a sign of cell toxicity.

### 3.4. Cell Viability and Bio-TEM Measurements

CeO_2_ NPs demonstrate selective cytotoxicity depending on the cell type and ability to uptake NPs and defend themselves from oxidative stress [45,48]. To determine the effects of the transition metal doping and the basic treatment (pH = 13.5) on the cytotoxicity of CeO_2_ NPs, we exposed HaCaT cells to CeO_2_, Cr@CeO_x_, and Fe@CeO_x_ NPs. In detail, while both Cr and Fe are known to induce intracellular ROS generation in various cellular contexts, Cr has also been shown to accumulate intracellular ROS by damaging cellular responses to oxidative stresses and to display genotoxic effect on cells by forming a DNA adduct. Of interest, our data demonstrating the localization of Cr@CeOx that mainly presides in cytoplasm suggest that a highly toxic effect of Cr@CeOx is likely related to its ability to accumulate intracellular ROS and to disrupt cellular homeostasis, which can be attributed to a distinctive electronic structure of Cr@CeOx.

The effects of these NPs on cell viability were then assessed using MTT (Figure 4a) and NRU assays (Figure 4b). Our choice of using HaCaT cells (derived from normal adult skin cells) is derived from the fact that skin is one of the first tissues that are exposed to NPs. The MTT and NRU assays revealed that exposure to undoped CeO_2_ and Fe@CeO_x_ NPs had no significant effect on the viability of the HaCaT cells up to 72 h, even at 100 μg/mL. Conversely, the Cr@CeO_x_ NPs displayed considerable cytotoxicity. Cell viability decreased by ~50% (~20%) during the MTT (NRU) assay after treatment with 100 μg/mL of NPs for 72 h (Figure 4a,b).

The STXM results shown in Figure 3 suggest that neither of the two types of NPs were able to penetrate the nucleus. However, the spatial resolution of STXM was limited to ~30 nm and employed photon energy. Thus, we could not determine the exact position of the penetrated NPs into the HaCaT cells because many components in the cell are composed of carbon and oxygen. Therefore, the STXM results should be confirmed by obtaining more detailed bio-TEM images of the same cells in the same position, which was marked on the TEM grid. Figure 4c,d show the bio-TEM images of the Cr@CeO_x_ and Fe@CeO_x_ NPs, respectively. Both types of NPs penetrated the cells and were located near their centers, but did not penetrate the nucleus (red color: NPs). In the bio-TEM images, the NP tracking technique was used to confirm that the red-colored NPs were CeO_2_-modified NPs. As shown in Appendix A, the difference between the particle-free bio-TEM image and the NP-transmitted TEM images can be clearly seen. As expected from the STXM data, both permeated NPs appear to easily pass through the cell membrane with no evidence of NP penetration at the nucleus located at the center of the cell. Based on the boundary of the nucleus, it was confirmed that all the nanoparticles exist outside the nucleus. By this experiment, we can confirm the selective improvement of the photocatalytic properties using double modifications (metal doping and pH treatment). Then we track the difference in cytotoxicity and the cell penetration of the modified CeO_2_ NPs.

## 4. Conclusions

In conclusion, we developed TM@CeO_x_ NPs characterized by high photocatalytic activity and no (or only minimal) toxic effects. It was confirmed that the internalization of the tested NPs did not considerably change the morphology and biochemical activities of cells among different tissue origins. Furthermore, while both Cr@CeO_x_ and Fe@CeO_x_ NPs showed enhanced photocatalytic activities, as compared to other NPs employed in multiple reactions, Cr@CeO_x_ NPs also showed severe cytotoxicity. Therefore, Cr@CeO_x_ NPs should be used with caution. Our study further demonstrated that Fe@CeO_x_ NPs can facilitate the control of reactions relevant to biomedical applications without severe negative effects on human cells. Meanwhile, Cr@CeO_x_ NPs have remarkable negative effects despite their good performance as photocatalysts. The NPs can reach the cell nuclei, but do not penetrate the nucleus (located at the center of the cells), and, hence, cannot alter the cell DNA information. Although limited, these results can be considered an efficient use of the tested NPs. The approach applied in this study to ascertain the photocatalytic properties and cell toxicity of the NPs can be extended to practical industrial applications.

## Figures and Tables

**Figure 1 nanomaterials-10-01543-f001:**
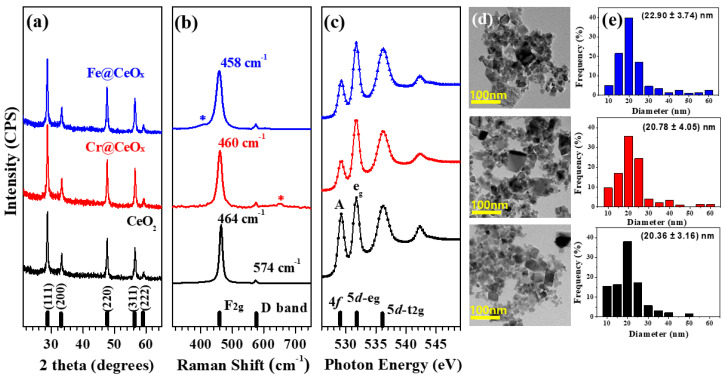
(**a**) XRD, (**b**) Raman, and (**c**) XAS spectra of CeO_2_ (black), Cr@CeO_x_ (red), and Fe@CeO_x_ nanoparticles (NPs) (blue color), and (**d**) their corresponding TEM images. (**e**) The histogram represents the size distribution of the nanoparticles measured by TEM images.

**Figure 2 nanomaterials-10-01543-f002:**
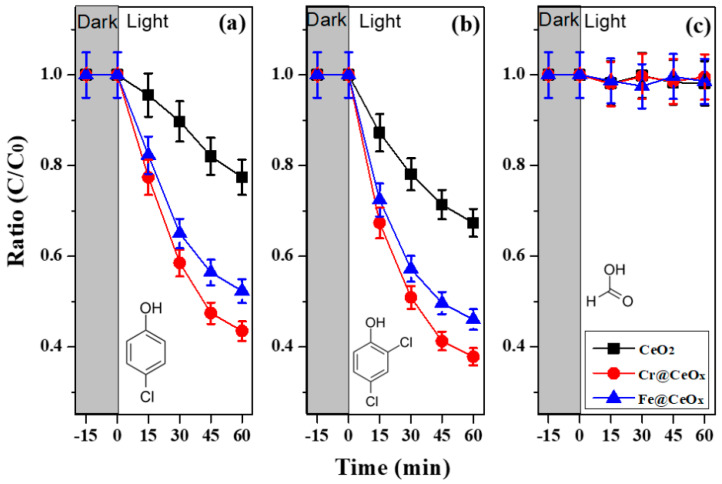
Photocatalytic degradation of (**a**) 4-CP, (**b**) 2,4-DCP, and (**c**) formic acid obtained by using CeO_2_, Cr@CeO_x_, and Fe@CeO_x_ NPs prepared at pH = 13.5 under ultraviolet (UV) irradiation. The experimental conditions were as follows: [catalyst] = 0.5 g/L, λ ≥ 320 nm, [4-CP]_0_ = 50 µM for a, [2,4-DCP]_0_ = 50 µM for b, and [formic acid]_0_ = 50 µM for c.

**Figure 3 nanomaterials-10-01543-f003:**
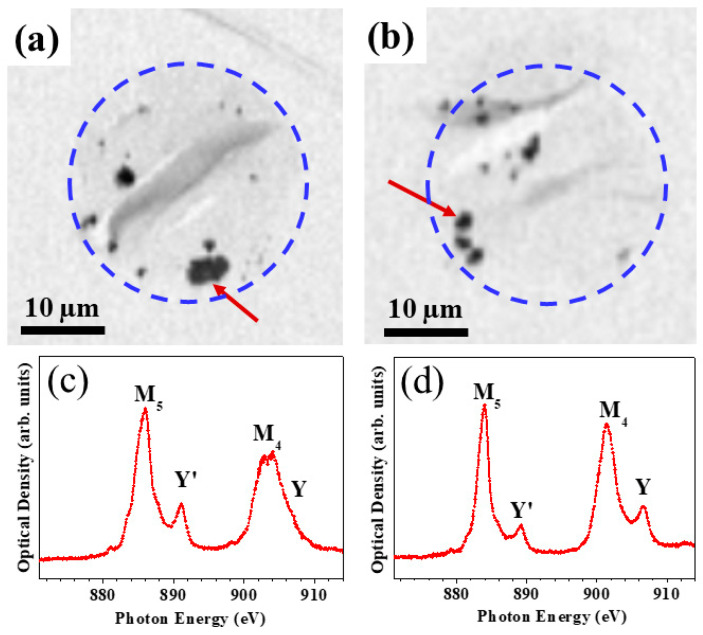
STXM images (**a**,**b**) and the XAS data (**c**,**d**) obtained for Cr@CeO_x_ nanoparticles (NPs) (**a**,**c**) and Fe@CeO_x_ NPs, which are shown in black (see the arrow position).

**Figure 4 nanomaterials-10-01543-f004:**
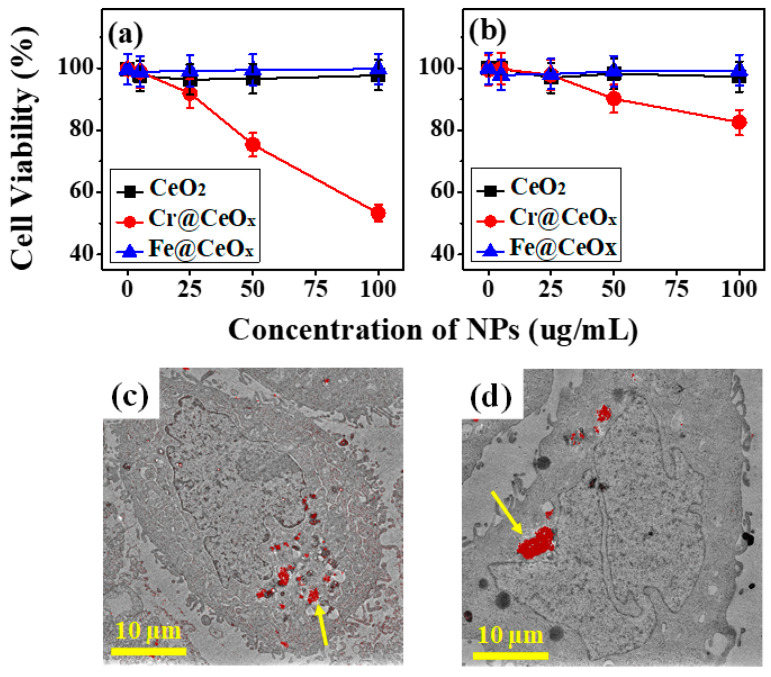
Quantification of the cytotoxicity of the nanoparticles (NPs) using MTT and NRU assays. HaCaT cells were treated with CeO_2_, Cr@CeO_x_, and Fe@CeO_x_ NPs at concentrations of 5 and 100 μg/mL for 72 h. (**a**) MTT assay and (**b**) NRU assay. Bio-TEM data (red color: NPs). Cr@CeO_x_ NPs are shown in (**c**), while Fe@CeO_x_ NPs are shown in (**d**). Arrow indicates the position of nanoparticles.

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
