# Peer review of "Comparison of Enhanced Photocatalytic Degradation Efficiency and Toxicity Evaluations of CeO2 Nanoparticles Synthesized Through Double-Modulation"

_nanomaterials, 2020, doi:10.3390/nano10081543_

Round 1

Reviewer 1 Report

The manuscript is mainly related to the photocatalytic enhanced performance of Fe/Cr doped and pH-modified CeO2 Nanoparticles. Some very preliminary results  regarding their toxicity are also reported.

The approach is undoubtedly interesting, but the results regarding toxicity are poor and their 'explanation' is mainly based on the sizes of the NPs, the evaluation of which is unfortunately not convincing.

At page 4, line 161, the authors wrote: "These TEM images revealed that all the NPs possessed diameters of 25–60 nm (average size is ~30 nm)".

The reported TEM images are, at a first glance, not in agreement with the authors' claim on the average size, that is the basis for the successive interpretation of toxicity. The size distribution, as resulting from TEM analysis, must be included in the revised version of the manuscript. Moreover, a careful analysis of the broadening of XRD peaks could give some further support to a more convincing determination of NPs distribution and average size.

For what concerns the STXM analysis, is not clear the rationale of its use in the economy of the paper, and the correlation of STXM and TEM data is poorly discussed.

In such a framework, my recommendation is for a major revision of the manuscript addressing carefully the above reported weaknesses.

Author Response

The manuscript reports fabrication of Cr or Fe doped CeO2, their use in photocatalysis and toxicity evaluation. The manuscript can be published after the following points are addressed:

First of all, we would like to appreciate the review’s comments for our work and valuable suggestions that will certainly improve it. Questions raised by the reviewer were answered point-by-point as follows.

Thank you for your helpful comment.

We hope your understanding!

Sincerely yours,

Hangil Lee

Reviewer 2 Report

The manuscript reports fabrication of Cr or Fe doped CeO2, their use in photocatalysis and toxicity evaluation. The manuscript can be published after the following points are addressed:

  1. The authors should describe the experimental in more detail: CeO2 synthesis, chemical for pH treatment, aqueous solution, photocatalytic experiments (amount of particles, volume of solution, power density of light source, irradiation area and stirring speed).
  2. More characterizations are required: XPS to examine doping, UV-vis, bandgap evaluation.
  3. What is the contribution of metal oxides (Fe2O3, Cr2O3) in both photocatalytic activity and toxicity?
  4. Why did the authors choose that doping concentration?
  5. The ratio between the second and third peak in XAS indicates the amount of vacancy (ACS Omega 2017, 2, 6, 2544–2551). Do the authors have any comments on this?

Author Response

(The authors gave the same response as above.)

Round 2

Reviewer 1 Report

The authors made some work of revision, but its 'accuracy' and deep can be improved.

Considering the relevance of NPs size in the economy of the paper and the 'brutal' changes made by teh authors in the reported values, a further revision is mandatorily recommended in order to make fully convincing the authors' claims on this fundamental issue.

In particular, I wrote in my previous report: <Moreover, a careful analysis of the broadening of XRD peaks could give some further support to a more convincing determination of NPs distribution and average size.> The authors simply overlapped the different peaks without making an evaluation of the crystallites sizes on the basis o f the detected broadening, using Scherrer or other alternative methods. This is necessary in order to confirm, on a larger number of crystallites, the information achieved by TEM, on a very local area and on a limited number of crystallites.

The authors also wrote, making possible the existence of negative sizes...:

Fe@CeOx (22.90 ± 37.42 nm)> Cr@CeOx (20.78 ± 40.47 nm)> CeO2 (20.36 ± 31.64 nm) NPs. 

The description of the revealed distribution must be clearly reported in a different way

Finally, the useful of STXM in the economy of the paper remain unclear. It is not clear where and what is the complementary relationship of this analysis with TEM characterization, able alone to reveal in a very precise and unambiguous way the position of nanoparticles penetrating the cell. Is it really necessary to include STXM analysis in the manuscript without to add nothing more on the achieved information?

Author Response

The authors made some work of revision, but its 'accuracy' and deep can be improved. Considering the relevance of NPs size in the economy of the paper and the 'brutal' changes made by the authors in the reported values, a further revision is mandatorily recommended in order to make fully convincing the authors' claims on this fundamental issue.

First of all, we are sorry for not concerning your helpful comment in the previous reply letter. In the 2nd reply letter, we certainly tried to improve it. Questions raised by the reviewer were answered point-by-point as follows.

yours sincerely,

Hangil lee

please check the attached file.

Reviewer 2 Report

The authors have revised the manuscript and I now recommend accepting the manuscript.

Author Response

Dear Sir 

Thank you for your decision.

Yours sincerely,

Hangil Lee